# Shoulder Injury Related to Vaccine Administration (SIRVA) Is Real: A Case Report

**DOI:** 10.3390/vaccines11071164

**Published:** 2023-06-27

**Authors:** Laura Jane Mackenzie, Jaquelin Anne Bousie, Mary-Jessimine Ann Bushell, Phillip Newman

**Affiliations:** 1Faculty of Health (Physiotherapy), University of Canberra, Canberra 2617, Australia; 2Faculty of Health (Pharmacy), University of Canberra, Canberra 2617, Australia; 3UC Research Institute for Sport and Exercise, Canberra 2617, Australia

**Keywords:** SIRVA, adhesive capsulitis, case report, frozen shoulder, vaccination

## Abstract

This study presents a case of SIRVA-induced adhesive capsulitis and the subsequent physiotherapy intervention. It details the patient’s journey using CARE guidelines. The main symptoms included persistent pain and a reduced range of motion for flexion, abduction, and internal and external rotation of the shoulder. Interventions included active and passive mobilisation via capsular stretching, and home exercise programs. At more than two years post-injury, the patient has ongoing pain, restricted shoulder movement, and disability. This highlights the importance of healthcare practitioners’ knowledge of SIRVA. Vaccinating practitioners should be aware of the mechanism of injury of SIRVA for preventing such injuries. First-contact practitioners should be aware of SIRVA-induced conditions to ensure timely and correct diagnosis and management of SIRVA-induced conditions.

## 1. Introduction

Despite the clear benefits of vaccines, their side effects have been under intense scrutiny by the public and the media. However, an often overlooked aspect of vaccination is the administration process and the poorly understood iatrogenic injuries that may result from improper vaccine administration techniques. Shoulder injury related to vaccine administration (SIRVA) is an adverse event following immunisation (AEFI) due to incorrect administration of a vaccine into the surrounding structures of the shoulder rather than the targeted deltoid muscle bulk [1]. Awareness of SIRVA amongst vaccinators and first-contact clinicians is poor, with varying definitions resulting in inconsistent reporting, unclear prevalence, and a poor understanding of its management [1]. Australian criteria for SIRVA include symptom onset within 24–48 h of a vaccination, pain on movement, restricted range of motion to the affected limb, abnormalities on medical imaging, and/or suspicion of incorrect vaccine administration technique [1,2]. The authors propose that to be considered a SIRVA, symptoms must have no response to over-the-counter analgesics (e.g., paracetamol, non-steroidal anti-inflammatory drugs (NSAIDs)) and last longer than a standard vaccination response of approximately one week [3]. SIRVA is considered by the Australian Department of Health and Aged Care and the Institute of Safe Medication Practices to be preventable through the use of proper landmarking, combined with a comprehensive understanding of the underlying anatomy [4,5].

Whilst the term SIRVA refers to a shoulder-related AEFI, it has been used to describe a range of conditions involving several anatomical structures of the shoulder. Known SIRVA-induced conditions are presented in Box 1 [1,4]. One of the key differentiations between SIRVA and other adverse reactions following vaccination is the iatrogenic nature of the syndrome. Vaccines may be delivered too high into the shoulder (glenohumeral) joint and miss the deltoid muscle’s bulk, resulting in the vaccine being injected into the shoulder capsule [3]. Vaccines may also damage the axillary or radial nerves when delivered laterally or too low [6]. Damage to both the capsule and nerves can result from direct traumatic damage caused by the needle itself or an inflammatory response stemming from a localised reaction to the vaccine [1,7]. Due to the differing sites of potential injection, SIRVA-induced conditions can vary among patients.

Box 1Reported SIRVA-induced conditions [1,3,4,5,6,7,8,9,10].Adhesive capsulitis (frozen shoulder)ArthralgiaAxillary nerve palsyNeuritisOsteonecrosis of the humeral headRadial nerve palsyRotator cuff tearsRotator cuff tendinopathyShoulder impingement syndromeSubdeltoid/subacromial bursitisSynovitis

While SIRVA is considered to be very rare, to date, no examination of its incidence has been performed and, as such, the total number of cases remains unknown. However, the proportion of SIRVA reports from the total number of AEFI reports has been determined using flu vaccination data from 2010–2017 [8]. This study, completed using the Vaccine Adverse Event Reporting Scheme (VAERS) in the United States of America, found that 1.5–2.5% of reports were considered to be SIRVA, depending on the flu season [8]. Given the scarcity of reported cases, the majority of the literature pertains to its prevention and diagnosis, with little exploring specific treatment modalities, especially with regard to physiotherapy interventions, which are blanketed under “physiotherapy”.

We present the patient’s journey and the impact of SIRVA-induced adhesive capsulitis of the left shoulder following an influenza vaccine in 2020. The clinical history, clinical assessment findings, patient-reported outcome measures, and treatment modalities are reported. 

## 2. Case Presentation

### 2.1. Patient Details

In April of 2020, a 50-year-old female nurse with no previous left shoulder conditions was administered a quadrivalent influenza vaccine into her left (non-dominant) arm via a single injection into the anterolateral portion of the deltoid muscle. The patient described the location as being “higher than usual” at 2–3 cm below the acromion process and more lateral than the standard vaccination administration positions. The patient described pain on administration and an expected red “lump” and soreness in the week following, progressing to radiating pain to the forearm by two weeks post-vaccination. No improvement was seen for the following two weeks before further deterioration. Her initial concerns were dismissed by her general practitioner (GP), with whom she had first contact, who did not believe injuries following injections were possible. At approximately six weeks post-vaccination, the patient was experiencing an 8/10 pain score on shoulder flexion, abduction, and external rotation. Ultrasound-guided hydrocortisone and local anaesthetic injection (HCLA) was performed at this time with a diagnosis of bursitis, with no change in or resolution of her symptoms. Further symptoms noted during this acute stage included a reduced range of motion, pain, sleep disturbance, functional decline, reduced independence, and low mood. At approximately two months post-vaccination, the patient sought care from a work physiotherapist, who diagnosed SIRVA-induced adhesive capsulitis. A pharmacological intervention was undertaken pro re nata (PRN) for approximately 12 months. The patient continued to work full-time throughout the duration of this injury, due to COVID-19-related staffing shortage impacts, with an altered workload to avoid overhead positions or lifting, and was assisted in donning and doffing personal protective equipment (PPE) by her colleagues. During this time, the patient was also responsible for running a household with young children. Dressing and undressing were performed by her partner or children for the initial six months, as the patient was unable to perform this herself. The patient was in good health and physical fitness otherwise, with no other health conditions. The patient had experienced right shoulder adhesive capsulitis approximately eight years prior, which was treated with unsuccessful hydro-dilation and self-management, self-resolving in approximately 18 months. However, she had no medical conditions of risk, such as diabetes or thyroid conditions.

### 2.2. Clinical Findings

At the initial assessment by the treating physiotherapist (20 months post-vaccination), the patient had significant pain and a reduction in active and passive range of motion (ROM) (Figure 1; Table A1, Appendix A). Muscle strength was tested using dynamometry for the rotator cuff muscles, and was significantly reduced (Table A1, Appendix A). The shoulder musculature was noted as visibly wasted by the treating physiotherapist, particularly at the deltoid muscle. No neurological symptoms or reduced sensation over the deltoid region were found on assessment. The cervical spine and thoracic spine were free from restriction, and no associated or referred pain was noted.

### 2.3. Timeline

Following initial vaccination, the patient experienced an initial freezing stage of approximately 3 months in which she underwent pharmacological intervention including US guided HCLA injection. During the frozen stage, lasting approximately 18 months, pharmacology intervention was utilised to manage pain. Once thawing, demonstrated by a gradual increase in range of motion, was noted physiotherpay intervention was reintroduced. A visual timeline of the patient experience, including timing of pharmacological and physiotherpay interventions, is presented in Figure 2.

### 2.4. Diagnostic Assessment

The initial assessment by the general practitioner included an ultrasound with an option for guided HCLA injection. The ultrasound findings showed thickening and impingement of the subacromial bursa, indicative of left subacromial bursitis. As a result, ultrasound-guided HCLA using a mixture of 5 cc of 2% lignocaine and 1 cc of Celestone was performed with no immediate complications. Self-reported survey data using the Disabilities of the Arm, Shoulder and Hand (DASH) outcome measure were then collected in retrospect for the initial injury and scored 71/100, with a higher score indicating higher levels of disability [11].

### 2.5. Differential Diagnosis

SIRVA can result in several shoulder pathologies, depending on the structure injured during the administration [1]. Pathologies include the initial needlestick penetration, primary inflammatory responses occurring when a structure is directly injected, chemical irritation from proximity to antigens, or secondary inflammation from the surrounding tissues [1]. In this instance, the occurrence of SIRVA, in which the vaccine was administered into the shoulder capsule, conceivably triggered a pathological cascade of primary inflammation and fibrotic changes leading to adhesive capsulitis. Differential diagnoses considered included bursitis and rotator cuff pathology, though both were ruled out due to the clinical presentation, the progressive worsening of ROM, and the lack of response to anti-inflammatory pharmacology [12].

Adhesive capsulitis, or frozen shoulder, is defined by a global restriction of ROM in both active and passive movements in at least two planes [12]. According to the clinical findings (Table A1, Appendix A), the patient fitted the accepted clinical definition for a global restriction of range of movement (ROM), that is, external rotation < 10°, internal rotation < L5, and flexion < 100° [12]. Adhesive capsulitis is typically described as following three pathological stages of freezing, frozen, and thawing [8]. These stages are characterised by 2–6 months of pain and reducing ROM, 4–12 months of global restriction secondary to the adhesions, and 6–24 months of gradual increases in ROM [12]. While the stages are extremely patient-dependent, typical cases of adhesive capsulitis resolve within three years; however, long-term follow-ups have shown residual mild to severe symptoms in 40% of sufferers after five years [13].

### 2.6. Treatment

The pharmacological intervention included (prescribed) meloxicam and over-the-counter analgesics (paracetamol and NSAIDs). These were taken on a PRN basis throughout the painful stage of the adhesive capsulitis (approximately 0–12 months). An injection of hydrocortisone and a local anaesthetic was administered at approximately one month after vaccination with nil effect. The physiotherapy intervention was initiated in January 2022, as pain and stiffness did not allow for an earlier intervention. The physiotherapy treatment utilised included passive physiological mobilisations performed as Grade III sustained holds in progressive end-range positions for abduction, external rotation, and flexion. As these positions are at the end range and aim to disrupt the formed adhesions, the patient often experienced pain scores of 8–9/10 during these sessions. Soft tissue massage was also utilised to reduce tension in the periscapular muscles. The patient additionally performed a progressive home exercise program specific to the deficits noted from her adhesive capsulitis presentation (Table A2, Appendix A). The exercises included stretching and active assisted strengthening to target the movements of flexion, external rotation, and abduction.

### 2.7. Outcome and Follow-Up

At six months from the onset of the physiotherapy intervention, the patient had reduced pain and improved ROM (Table A1, Appendix A), but still only 22.2–70% of her normal expected range. Dynamometry reassessment of the rotator cuff muscles demonstrated an improvement in strength for the rotator cuff muscles; however, less than 90% limb symmetry remained (Table A1, Appendix A). Functional improvement was most significant for the patient, who was finally able to tie up her own hair at over two years post-vaccination. The follow-up DASH self-report scored 35/100, a reduction of 36 points, with the minimally clinically important difference for the DASH scale being 10.81–15 points [11]. While the patient had experienced a considerable improvement, significant disability was still noted, with a score of 0/100 indicating nil disability [11]. Further follow-up was not available, due to the patient’s commitments leading to the in-person physiotherapy being ceased.

## 3. Discussion

This study presents a physiotherapy perspective of the assessment and treatment of a SIRVA-induced adhesive capsulitis case, providing insight into the role that physiotherapy may play in shoulder conditions with this mechanism of injury. It is important that the first-contact practitioners be aware of SIRVA as an AEFI when patients present with shoulder pain post-vaccination. This patient was, unfortunately, dismissed by her first-contact practitioner, who appeared to be unaware of SIRVA and consequently treated her pharmacologically for bursitis, resulting in a delayed referral for physiotherapy. Blanket treatments such as corticosteroids have been suggested as first-line treatments for SIRVA. While corticosteroids may be effective in the treatment of some SIRVA-induced conditions such as bursitis or adhesive capsulitis [14], they are likely to be ineffective or detrimental for other conditions such as infective bursitis, tendinopathy, or osteonecrosis of the humeral head [1]. As there are a range of conditions that may result from SIRVA, it is important that a correct diagnosis is made following the adverse vaccination event, and that the corresponding treatments are individualised to the patient’s specific clinical presentation and diagnosis.

There have been prior links to SIRVA resulting in adhesive capsulitis in case studies, case series, and large-scale retrospective cohort studies, many of which had little to no known risk factors for its development [8,9,10]. In all cases, the underlying mechanism of improper injection leading to SIRVA was noted. Due to the patient being middle aged and female, and having had a prior incidence of adhesive capsulitis, she may have had an increased risk for its development, in addition to other known SIRVA-induced conditions such as bursitis and rotator cuff tendinopathy [1,8]. However, the onset of pain from the minute of injection, which continued to escalate, leaves little room for other causes to be viably considered. The trauma to the shoulder capsule that likely occurred during the improper vaccination (SIRVA) appears to have triggered a dormant risk for adhesive capsulitis and provided the innocuous trauma that sparked the inflammatory cascade. In this case, the trauma itself, and the subsequent SIRVA, was avoidable [1].

There are numerous factors that play a role in ensuring the safe delivery of vaccines. The upper arm should be completely exposed during vaccination into the deltoid muscle, as adequate exposure cannot be obtained when clothing is pulled down or rolled up, and increases the risk of misinjection [4,15,16,17]. Vaccinators are also taught to use anatomical landmarking techniques in order to determine a safe injecting zone. The *Australian Immunisation Handbook* recommends forming an inverted triangle through palpation of the acromion process and the deltoid tuberosity, with finger positions between these landmarks highlighting the safe zone. [4,15]. The landmarking technique described in the *Australian Immunisation Handbook* is demonstrated in Figure 3A,B. Other methods involve the practitioner’s fingers covering the subacromial/subdeltoid bursae, which can be found through palpation of the acromion process. This method is proposed to prevent the immunisation being delivered too high into the glenohumeral joint; however, it does not prevent the practitioner from delivering the immunisation too low [18,19,20,21]. While these and various other landmarking techniques are proposed as being effective for locating the thickest point of the deltoid muscle, it must be stated that there is limited evidence to support them or determine which methods are most effective [22].

Despite the use of landmarking techniques being imperative for safe vaccination practices, they are not always implemented. A small study (*n* = 76) by McGarvey and Hooper [23], using a self-reported questionnaire, found that only three (4%) vaccinators used landmarking techniques when administering vaccines [23]. The study also reported that vaccinators’ knowledge of the at-risk anatomy was low [23]. While this study was small and self-reporting questionnaires are subject to response bias and prone to overestimation of one’s abilities, it highlights that vaccinators do not uniformly use landmarking techniques. All respondents to the questionnaire were experienced practitioners in their fields [23]. However, the specific immunisation training reported was highly varied. All 32 nurse practitioners reported receiving immunisation and IM injection training prior to qualification [23]. Three nursing practitioners and six medical doctors reported undergoing further immunisation training after registration [23]. Alarmingly, 30 doctors self-reported as having no formal training, and four were unsure if they had received formal training [23]. A similar study examining healthcare practitioners’ knowledge of SIRVA, shoulder anatomy, and safe injecting was performed in 2022 [24]. Interestingly, non-immunising professions scored higher than immunising professions in knowledge of both the shoulder’s anatomy and safe injection (67% vs. 71% and 36% vs. 51%, respectively) [24]. The mean scores for combined groups were lowest for shoulder anatomy, with only 42% accuracy [24]. Concerningly, only 54% of authorised immunisers were able to correctly locate a 20 × 40 mm safe injecting zone on a standardised image [24]. When anatomical landmarking techniques are reliant on the practitioners’ knowledge of the underlying upper limb anatomy, it is concerning that knowledge levels are so low. The paucity of educational materials related to SIRVA is of concern. The available materials from the *Australian Immunisation Handbook* lack details related to the underlying anatomical structures, instead focussing only on landmarking techniques for prevention [4,15]. The consequences of incorrect administration techniques are poorly explained and give little insight into the long-term outcomes for patients [4,15]. Educational materials on SIRVA should address the relevant anatomical structures, strategies for prevention, the definition and diagnostic criteria for SIRVA, and the consequences and long-term outcomes for patients.

This case study highlights the experience of a patient suffering from SIRVA-induced adhesive capsulitis, and the long journey to recovery. Vaccinating healthcare professionals should be aware of the risks of incorrect vaccine administration techniques and the potential long-standing injuries patients may suffer because of errors during vaccination. Healthcare professionals who are first-contact practitioners treating shoulder pain must recognise the links among incorrect vaccine administration techniques, SIRVA, and SIRVA-induced conditions. This increased awareness and understanding of SIRVA-related conditions will ensure correct early diagnosis and enable such practitioners to treat their patients effectively and holistically.

Limitations: This study is limited by the retrospective nature of the data collection, and the patient’s prior history of adhesive capsulitis.

## 4. Conclusions

SIRVA is an AEFI that can have long-standing symptoms and disability for patients. Healthcare practitioners should be aware of SIRVA, SIRVA-induced conditions that may arise, and how to appropriately refer or manage patients. Physiotherapists are first-contact practitioners who may see and treat SIRVA-induced conditions. Physiotherapy and/or pharmaceuticals are appropriate interventions for SIRVA-induced adhesive capsulitis. This study documents a case of SIRVA-induced adhesive capsulitis and presents the management specific to that condition. SIRVA can induce many shoulder conditions, which should be diagnosed and managed individually and appropriately.

## Figures and Tables

**Figure 1 vaccines-11-01164-f001:**
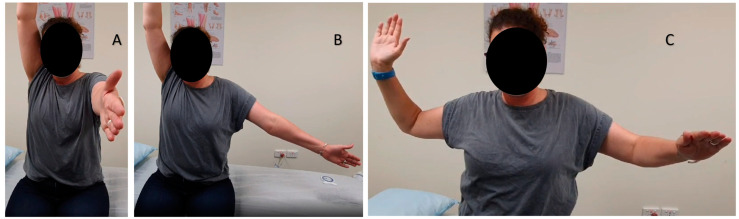
Range of motion deficits, right vs. left: (**A**) flexion; (**B**) abduction; (**C**) external rotation from 90° abduction.

**Figure 2 vaccines-11-01164-f002:**
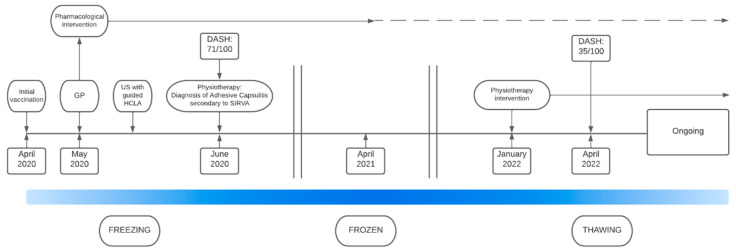
Timeline of the patient’s experience. Abbreviations: GP, general practitioner; US, ultrasound; HCLA, hydrocortisone and local anaesthetic injection; SIRVA, shoulder injury related to vaccine administration; DASH, disabilities of the arm, shoulder, and hand. A higher score indicates a higher level of disability; scores of 0 indicate nil disability.

**Figure 3 vaccines-11-01164-f003:**
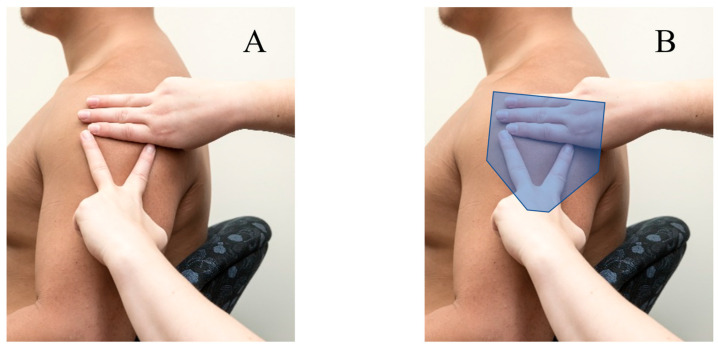
Anatomical landmarking example. (**A**) Demonstration of the *Australian Immunisation Handbook*’s landmarking technique. (**B**) Demonstration of the *Australian Immunisation Handbook*’s landmarking technique, including an overlay of the approximate shape and position of the deltoid muscle [1].

## Data Availability

All data relevant to the study are included in the article.

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
