# Peer review of "Shoulder Injury Related to Vaccine Administration (SIRVA) Is Real: A Case Report"

_vaccines, 2023, doi:10.3390/vaccines11071164_

Round 1

Reviewer 1 Report

Summary:

This is a case report of a single individual who reported experienced shoulder problems. These problems were traced back to a flu vaccine injection. The should issues appear to be resolving six months after the initiation of physical therapy treatment.

Suggestions:

The title should be shortened to ‘Shoulder Injury Related to Vaccine Administration (SIRVA) is real: A Case Report’. The information in the manuscript is NOT specific to only health professionals. It goes without saying that SIRVA impacts a patient’s life.

The last two paragraphs of the Introduction can be deleted without compromising the main point of this case report. These paragraphs appear to be a quoted narrative from a patient regarding the path from a flu vaccination through the difficulties in getting a physician to determine a diagnosis. It is unknown if this is the narrative from the patient in the case report or some other patient. In addition, this type of presentation is inappropriate for this type of journal article.

The Case Presentation states the patient sought care from a physiotherapist two months post-vaccination who diagnosed SIRVA. Pharmacological interventions were attempted for the following 12 months (a total of 14 months). It is surprising the physiotherapist did not collect initial performance and strength data at the initial visit, two months after vaccination at the time of diagnosis.

Figure 1 can be deleted as the data are included in Appendix Table 1.

Appendix Table 2 (Exercise program one), Appendix Table 3 (Exercise program two), and Appendix TABLE 1 (Exercise program three, apparently a numbering topographical error) can be deleted from the manuscript. The information in these tables, if necessary, can be included into one or two short paragraphs to the Treatment section of the manuscript.

The authors do not state why follow-up on this patient was stopped after 6 months of physiotherapy. Perhaps greater strength and range of motion toward ‘normal’ would be observed at 9 or 12 months?

Given the worldwide increase in vaccinations as a result of the COVID-19 pandemic over the past few years, it would be interesting if the authors presented data, or addressed, if data are not available, the incidences of SIRVA. Even though SIRVA is rare, would the number of cases have been expected to increase?

Reviewer 2 Report

Thank you for the opportunity to review this report about a case of SIRVA following an immunisation. Getting a case report published is always a challenge because, given the simplicity of the type of article, it is important to present elements that make the content nonetheless meaningful. This case report has potential but the authors should improve some aspects of the presentation. Below are my specific comments.

1. The Introduction can be improved: more emphasis needs to be placed on presenting this specific case, and specify what the purpose of the presentation is, what goals the authors propose in reporting the case, the literature gap, what other reports are present, etc. I would personally omit the patient's first-person narrative. The information gathered can be reported more rigorously and less colloquially.

2. The presentation of the case is very good, well articulated and detailed and with useful supports of tables and photographs. I would advise the authors to reorganize all headings up to and including #8, organizing the whole section as a 'case' and proceeding with sub-headings.

3. I find the Discussion to be too unbalanced on physiotherapy content and considerations. It is equally important to use the case presentation for the prevention of the event, i.e., educating practitioners on proper inoculation. Considerations must be developped in this sense. I would also recommend enriching the literature cited and discussed. Same applies to the Conclusions.

Round 2

Reviewer 2 Report

The authors have answered all the comments and changed the manuscript accordingly. Although this remains a simple design article, it has been improved in terms of structure and contents.